# Listening to Self-Selected Music during Warm-Up Improves Anaerobic Performance through Enhancement of the Excitability of the Cerebral Cortex

**Shiyan Zhang** [1,2,†], **Juan Yang** [2,†], **Xifeng Tao** [2], **Liwen Du** [2], **Xiang Li** [2], **Yuanyuan Lv** [1,3], **Xiao Hou** [1,4,*] and **Laikang Yu** [1,2,*]

1   Key Laboratory of Physical Fitness and Exercise, Ministry of Education, Beijing Sport University, Beijing 100084, China; 17634977464@163.com (S.Z.); sunflowerlyy@bsu.edu.cn (Y.L.)
2   Department of Sports Performance, Beijing Sport University, Beijing 100084, China; yj9264582022@163.com (J.Y.); txf19983480529@126.com (X.T.); duliwen2019@126.com (L.D.); jinaxun0889@yeah.net (X.L.)
3   China Institute of Sport and Health Science, Beijing Sport University, Beijing 100084, China
4   School of Sport Sciences, Beijing Sport University, Beijing 100084, China
*   Correspondence: houxiao0327@bsu.edu.cn (X.H.); yulaikang@126.com (L.Y.)
†   These authors contributed equally to this work.

**Abstract:** The study investigated the effects of listening to self-selected music during a warm-up on brain wave synchronization/desynchronization and Wingate test performance. Seventeen healthy young men were required to complete a 10 min warm-up session with or without music intervention, followed by an electroencephalogram (EEG) or Wingate test, respectively. The ratings of perceived exertion (RPE) and heart rate (HR) were recorded immediately after the Wingate test. Compared with no music intervention, listening to self-selected music during a warm-up significantly increased peak power and mean power in the Wingate test ($p < 0.05$), upregulated the α energy percentage in the F3, C3, P3, O1, T3, F4, and Fp2 regions ($p < 0.05$) and β energy percentage in the F3, O1, and T5 regions ($p < 0.05$), while it downregulated the δ energy percentage in the F3, P3, O1, F4, and F8 regions ($p < 0.05$), θ/β in the F3 and O1 regions ($p < 0.05$), and (θ+α)/(α+β) in the F3 region ($p < 0.05$). However, there were no significant differences in the minimum power and fatigue index in the Wingate test between the music intervention and no music intervention, or in RPE and HR after the Wingate test ($p > 0.05$). This study demonstrated that listening to self-selected music during a warm-up enhances cortical excitability by upregulating the α and β energy percentages and downregulating the δ energy percentage, which may represent a potential mechanism by which listening to self-selected music during a warm-up improves anaerobic performance in healthy young men.

**Keywords:** self-selected music; warm-up; electroencephalogram; Wingate; fatigue; anaerobic performance

## 1. Introduction

Music is a potential external stimulus that can motivate people to perform at their best during exercise. Listening to music during exercise may produce a more positive emotional value, improved physical performance, reduced perceived exertion, and increased physiological efficiency [1]. The beneficial effect of music on the practitioner is usually influenced by the following two aspects: on the one hand, the intensity and duration of exercise; on the other hand, the rhythm, melody, and volume of the music, as well as the practitioner's personal preference [2–4]. A previous study has shown that explosive movement and changes in the emotional state while listening self-selected music can improve acute power performance [5]. In addition, the timing of listening to music may also have an effect on this gain. Ballmann et al. [6] showed that listening to music before performing a bench press can

improve muscle explosive power and strength endurance, making it easier to operate in the actual training process. Listening to music before exercise may have a positive impact on short-term tasks, mainly anaerobic tasks, such as grip strength, Wingate test, etc. [7].

Eliakim et al. [8] conducted a music intervention during a warm-up session in 24 national-level elite young volleyball players of different genders. The results showed that although music had no significant effect on the mean anaerobic output or fatigue index in both sexes, it may have a short-term beneficial effect on anaerobic performance, which was consistent with the result of Meglic et al. [9], showing that listening to self-selected warm-up music had an ergogenic effect during repeated sprints and improved motivation to exercise. In addition, the result of a systematic review suggested that listening to music during the Wingate test might physiologically enhance relative anaerobic exercise performance [10]. Therefore, listening to warm-up music prior to high-intensity exercise may aid in optimizing anaerobic performance, although the reasons remain speculative. A previous study suggested that the beneficial effect of music on short-term anaerobic tasks may be related to increased motivation to exercise and decreased ratings of perceived exertion (RPE) [2]. Furthermore, Lane et al. [11] compared the effects of self-selected music intervention and Audiofuel music intervention on changes in emotional state and performance during running. Participants in both groups reported an increase in pleasant emotions, a decrease in unpleasant emotions, and a significant improvement in performance following intervention. Further analysis showed that higher levels of arousal were associated with better performance. Potential reasons for the enhanced effect of music on performance include a reduced perception of fatigue, increased exercise satisfaction, and the advantage of synchronized rhythms [12–14].

Recent studies have shown that music has a significant effect on brain electrical activity, which may contribute to improved performance after music interventions [15,16]. In addition, music therapy has a wide range of applications in the field of nursing and rehabilitation, providing sufficient evidence for the impact of music intervention on brain neuroimaging. A previous study showed that listening to music had a positive effect on the recovery of language function after stroke, and the increased activation of the left frontal aslant tract mediated structural neuroplastic changes in the language network [17]. Brattico et al. [18] used functional magnetic resonance imaging (fMRI) to investigate brain responses to sad and positive music with or without lyrics. They found that sad music with lyrics recruited more brain activity in several regions, including the caudate nucleus, parahippocampal gyrus, inferior frontal gyrus, amygdala, precentral gyrus, putamen, insula, and auditory cortex. On the other hand, positive music with lyrics led to activity in the limbic system and certain regions of the right inferior frontal gyrus. Whitehead et al. [19] found differences in the responses to sound and music stimuli in different regions of the temporal lobe of the brain; specifically, the superior temporal sulcus and gyrus were more active in response to sound stimuli, whereas the planum polare and temporal were more active in response to music stimuli.

An electroencephalogram (EEG) records the electrical discharges of a group of neurons in the brain during synchronous activation through electrodes attached to the scalp [20,21]. EEG power reflects the number of neurons that electrically discharge synchronously [22]. When people are in different physiological states, such as being awake, sleepy, and asleep, the frequency and waveform of EEG signals change significantly. Depending on the frequency, EEG signals can be classified as alpha ($\alpha$), beta ($\beta$), theta ($\theta$), and delta ($\delta$) waves [23]. A previous study showed that the $\alpha$ wave displays more activation in intermittent noisy environments than in completely silent environments [24]. Music has an effect on athletic performance and fatigue recovery, which is also expressed in the brain waves. Moraes et al. [25] showed that the EEG spectrum may differ before and after acute exercise, including power changes in the $\alpha$ and $\beta$ ranges of the EEG spectrum. In addition, music can stimulate the human cerebral cortex and is closely related to the electrical activity of the human brain, as demonstrated by its effect on brain waves after exercise. Bigliassi et al. [15] showed that music downregulated $\theta$ waves in the frontal, central, and parietal regions of

the brain during exercise, which may be the brain mechanism by which music ameliorated fatigue-related symptoms during the execution of fatiguing motor tasks. Rearrangement of β waves after exercise may be associated with attentional demands and higher levels of arousal [25]. Bigliassi et al. [26] showed that the psychological effects of music on exercise could be associated with the upregulation of β waves in the frontal and frontal-central regions of the brain, which appears to elicit a more positive emotional state. To sum up, music can have direct and indirect effects on people's psychology and physiology. Therefore, listening to music before and during exercise may enhance athletic performance and produce neural responses based on changes in EEG.

In the present study, we aimed to investigate the effects of listening to self-selected music during a warm-up session on brain wave synchronization/desynchronization and Wingate test performance in healthy young men in order to explore the possible mechanisms by which music intervention improves anaerobic performance.

## 2. Materials and Methods

### 2.1. Study Design

This study included a randomized crossover design experiment in which subjects were asked to attend the laboratory five times. During the first visit, subjects determined self-selected music and were familiarized with the experimental procedures. During the other four visits, subjects were required to complete a 10 min warm-up session in one of two conditions (A: music intervention (M); B: no music intervention (NM)), followed by an EEG or Wingate test, respectively. RPE (Borg's scale) and heart rate (HR) were recorded immediately after the Wingate test. A 5-day washout period was included between interventions to control for residual effects.

### 2.2. Participants

Seventeen healthy young men (age: 21.31 ± 1.01 years; height: 179.35 ± 4.36 cm; weight: 74.65 ± 7.05 kg; body mass index (BMI): 23.18 ± 1.63 $kg/m^2$) volunteered to participate in this study. All subjects had no potential medical problems and no history of ankle, knee or back pathology that compromised their participation or performance in the study. None of the subjects were taking drugs or had antecedents of neurological or psychopathological conditions. All participants were fully aware of the procedures, possible risks, and purpose of the study, preceded by a 24 h period without tobacco, caffeine, alcohol, and vigorous exercise. The study protocol was approved by the Ethics Committee of Beijing Sport University (approval number: 2022115H), and all procedures were carried out in accordance with the recommendations of the Declaration of Helsinki. All participants gave written informed consent in accordance with the Declaration of Helsinki. The descriptive characteristics for the study subjects can be viewed in Table 1.

**Table 1.** Participants' physical characteristics (*n* = 17).

|  | Mean ± SD |
| --- | --- |
| Age (years) | 21.31 ± 1.01 |
| Height (cm) | 179.35 ± 4.36 |
| Weight (kg) | 74.65 ± 7.05 |
| Body mass index ($kg/m^2$) | 23.18 ± 1.63 |

### 2.3. Procedures

In this randomized crossover study, each subject completed five study visits. On the first visit, self-selected music of each subject was determined. On each of the other four visits, the subjects were required to arrive at the laboratory at least 2–3 h post-absorptive. Subjects adjusted the seat to the appropriate height before the warm-up and wore the same Bluetooth headsets (with or without music intervention) during the warm-up. Music was played using an iPhone (Apple Inc., Cupertino, CA, USA), with the volume controlled by

the subjects themselves. During the experiment, subjects were required to perform 10 min of relaxed riding with or without music intervention, with a load of 20% × 0.075 kg/body weight and a speed control of 75 r/min, followed by an EEG or Wingate test, respectively. In addition, RPE and HR were recorded immediately after the Wingate test. A 5-day washout period was included between the interventions to control for residual effects.

### 2.4. RPE

Borg's RPE scale was used to assess perceived exertion during the cycling warm-up, which provided a whole-body fatigue assessment and represented feedback from the respiratory, cardiovascular, and musculoskeletal systems. We recorded the RPE of subjects after the Wingate test.

### 2.5. HR

HR was measured via telemetry (RS800CX, Polar Electro, Kempele, Finland). The heart rate chest belt was worn on the subject's chest and the receiver was located to the left of the midline. The elastic band was adjusted so that the position of the receiver did not change during the ride. We recorded the HR of subjects after the Wingate test.

### 2.6. Wingate Test

Anaerobic performance was measured by the Wingate cycle test (Monark 894E, Vansbro, Sweden). At the beginning of the test, the subject pedaled at full strength without any resistance. Once the maximum speed was reached (usually after 3–4 s), a predetermined load (7.5% of body weight in kilograms) was applied and the subject was required to maintain the pedaling for 30 s. Subjects were asked to remain seated throughout the test. After the test, subjects were asked to perform a 2 min cool down. Peak power, mean power, minimum power, and fatigue index (FI) were measured and recorded during the test. Peak power, mean power, and minimum power were defined as the highest mechanical power output, average mechanical power output, and lowest mechanical power output recorded during the test, respectively. The FI was calculated by dividing the difference between the peak power and the minimum power by the peak power and then multiplying by 100 to determine the percentage [27].

### 2.7. EEG Recording

The EEG data were continuously sampled at 128 Hz from 16 active silver-silver chloride electrodes attached to an electrode cap (Brain Products Nation9128w, Shanghai, China), placed according to the international 10–20 system [28]. The 16 electrodes were placed on left frontal poel (Fp1), right frontal poel (Fp2), left frontal (F3), right frontal (F4), left central (C3), right central (C4), left parietal (P3), right parietal (P4), left occipital (O1), right occipital (O2), left anterior temporal (F7), right anterior temporal (F8), left mid-temporal (T3), right mid-temporal (T4), left posterior temporal (T5), and right posterior temporal (T6) regions, respectively. The reference electrodes were placed on the bilateral earlobes with interelectrode impedance kept below 5 kΩ to minimize noise artifacts.

During the EEG recording, subjects were asked to take a relaxed sitting position in the room with the light off, closing their eyes and avoiding any movement. After a 10 min warm-up with or without the music intervention, a 3 min EEG was recorded. Each dataset was visually checked for excessive noise. If more than 25% of the recording was determined to be noisy, the entire dataset was rejected. All EEG signals were preprocessed in MATLAB to remove artifacts and noise. Ocular and cardiac artifacts were removed using the independent components analysis (ICA) from the EEGLAB toolbox. Subsequently, all signals were digitally filtered with a band pass Butterworth filter between 0.5 and 50 Hz. For both tests, 30 s artifact-free data were selected for analysis, and the cortical EEG power spectrum energy and energy percentage were calculated. The power density was calculated for the following bands: $\delta$ (0.1–3 Hz), $\theta$ (4–8 Hz), $\alpha$ (8–12 Hz), and $\beta$ (12–28 Hz) bands [29]. In addition, we also calculated $\theta/\beta$ and $(\theta+\alpha)/(\alpha+\beta)$, which are related to fatigue [30,31].

*2.8. Statistical Analyses*

Data were presented as the mean $\pm$ standard deviation (SD). Normality was checked using the Shapiro–Wilk test. After confirming the normality of the data, a paired samples t-test was used. For non-normal data, non-parametric tests were used. In addition, Cohen's *d* was used as the effect size (ES) estimation with its strength being interpreted as the following: trivial (0–0.2), small (0.2–0.6), moderate (0.6–1.2), and large (1.2–2.0), and very large (>2.0) [32]. The significance level was set at $p < 0.05$. All statistical analyses were performed using SPSS version 26.0 (IBM, Armonk, NY, USA).

## 3. Results

*3.1. Effects of Self-Selected Music on Wingate Test Performance*

The effects of self-selected music on Wingate test performance are shown in Figure 1. Compared with no music intervention, listening to self-selected music during the warm-up significantly increased peak power (M: $888.20 \pm 170.95$ W, NM: $825.90 \pm 181.48$ W; Cohen's $d = 0.35$; 95% CI: 8.45 to 102.01; $t_{16} = 2.503$, $p = 0.024$) and mean power (M: $629.06 \pm 87.81$ W, NM: $589.2 \pm 99.33$ W; Cohen's $d = 0.42$; 95% CI: 9.38 to 61.10; $t_{16} = 2.889$, $p = 0.011$), while there were no significant effects on minimum power (M: $415.17 \pm 59.16$ W, NM: $390.63 \pm 63.68$ W; Cohen's $d = 0.39$; 95% CI: $-7.20$ to 56.29; $t_{16} = 1.639$, $p = 0.121$) and fatigue index (M: $51.67 \pm 11.23$, NM: $50.67 \pm 11.79$; Cohen's $d = 0.08$; 95% CI: $-4.9$ to 4.75; $t_{16} = 0.046$, $p = 0.964$) in the Wingate test.

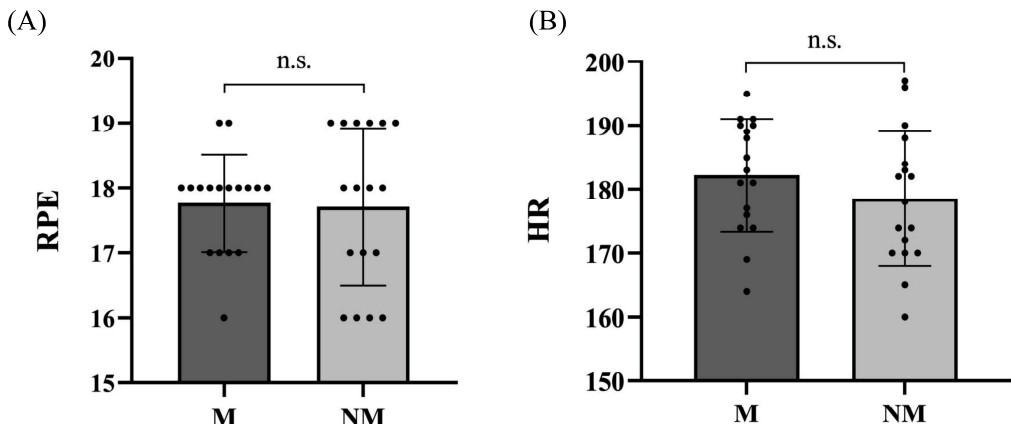

**Figure 1.** Listening to self-selected music during the warm-up had no significant effect on RPE ($p = 0.854$, (**A**)) and HR ($p = 0.222$, (**B**)) after the Wingate test. RPE, rating of perceived exertion; HR, heart rate; M, music intervention; NM, no music intervention; n.s., no significance. The dots represent the raw data.

*3.2. Effects of Self-Selected Music on RPE and HR*

Figure 2 illustrates the RPE and HR values recorded after the Wingate test. There were no significant differences in RPE (M: $17.76 \pm 0.75$, NM: $17.71 \pm 1.21$; Cohen's $d = 0.06$; 95% CI: $-0.60$ to 0.72; $t_{16} = 0.187$, $p = 0.854$) and HR (M: $182.24 \pm 8.79$, NM: $178.53 \pm 10.51$; Cohen's $d = 0.38$; 95% CI: $-2.44$ to 9.88; $t_{16} = 1.271$, $p = 0.222$) between the music intervention and no music intervention after the Wingate test.

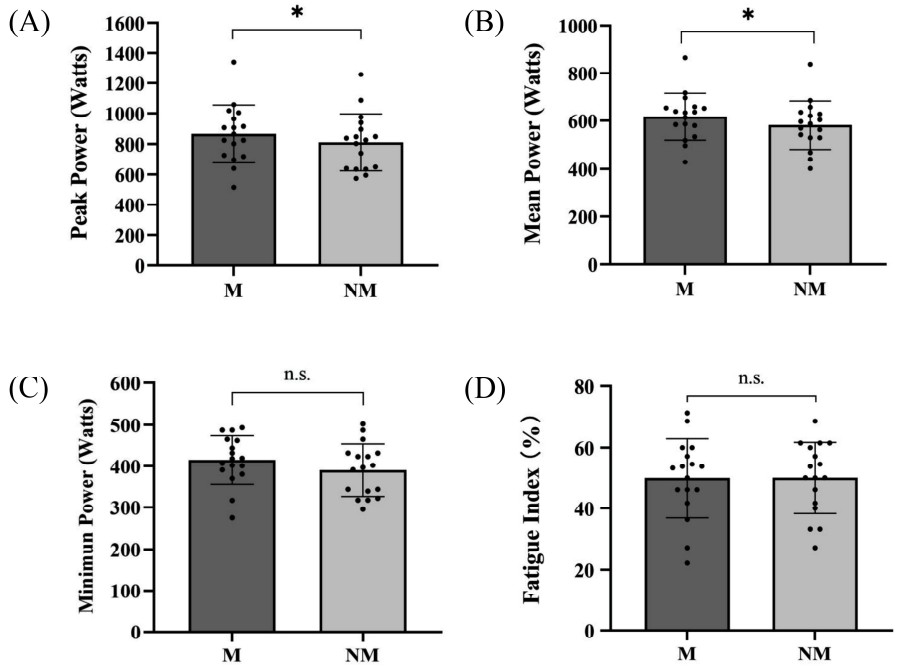

**Figure 2.** Listening to self-selected music during the warm-up significantly increased peak power
(* *p* = 0.024, (**A**)) and mean power (* *p* = 0.011, (**B**)), while there were no significant effects on minimum
power (*p* = 0.121, (**C**)) and fatigue index (*p* = 0.964, (**D**)) in the Wingate test. M, music intervention;
NM, no music intervention; n.s., no significance. The dots represent the raw data.

### 3.3. Effects of Self-Selected Music on EEG Signals

The effects of self-selected music on EEG signals are shown in Table 2. Compared with
no music intervention, listening to self-selected music during the warm-up significantly
upregulated the α energy percentage in the F3 (M: 24.55 ± 12.33%, NM: 16.41 ± 11.10%;
Cohen's *d* = 0.69; 95% CI: 0.91 to 15.35; $t_{16}$ = 2.389, *p* = 0.030), C3 (M: 28.72 ± 12.44%,
NM: 23.74 ± 14.77%; Cohen's *d* = 0.36; 95% CI: 0.27 to 9.68, $t_{16}$ = 2.245, *p* = 0.039), P3 (M:
31.74 ± 14.40%, NM: 19.49 ± 12.92%; Cohen's *d* = 0.89; 95% CI: 4.85 to 19.64, $t_{16}$ = 3.513,
*p* = 0.003), O1 (M: 30.95 ± 17.29%, NM: 18.35 ± 14.64%; Cohen's *d* = 0.78; 95% CI: 4.97 to
20.24, $t_{16}$ = 3.500, *p* = 0.003), T3 (M: 26.85 ± 12.93%, NM: 17.84 ± 9.26%; Cohen's *d* = 0.80;
95% CI: 2.68 to 15.33, $t_{16}$ = 3.019, *p* = 0.008), F4 (M: 32.40 ± 12.24%, NM: 24.78 ± 13.55%;
Cohen's *d* = 0.58; 95% CI: 0.28 to 14.94, $t_{16}$ = 2.204, *p* = 0.043), and Fp2 regions (M:
28.48 ± 13.36%, NM: 21.26 ± 11.76%; Cohen's *d* = 0.57; 95% CI: 0.76 to 13.67, $t_{16}$ = 2.371,
*p* = 0.031), β energy percentage in the F3 (M: 16.40 ± 11.81%, NM: 8.17 ± 5.89%; Co-
hen's *d* = 0.88; 95% CI: 2.50 to 13.95, $t_{16}$ = 3.045, *p* = 0.008), O1 (M: 11.25 ± 4.65%, NM:
6.29 ± 4.90%; Cohen's *d* = 1.03; 95% CI: 1.90 to 8.02, $t_{16}$ = 3.436, *p* = 0.003), and T5 regions
(M: 18.28 ± 8.35%, NM: 12.48 ± 7.49%; Cohen's *d* = 0.73; 95% CI: 0.45 to 11.14, $t_{16}$ = 2.300,
*p* = 0.035), while it downregulated the δ energy percentage in the F3 (M: 35.77 ± 13.97%,
NM: 51.27 ± 17.84%; Cohen's *d* = −0.96; 95% CI: 25.16 to 5.83, $t_{16}$ = −3.400, *p* = 0.004), P3 (M:
30.51 ± 10.22%, NM: 42.34 ± 19.78%; Cohen's *d* = −0.75; 95% CI: 22.33 to 1.32, $t_{16}$ = −2.387,
*p* = 0.030), O1 (M: 36.31 ± 18.35%, NM: 49.77 ± 16.80%; Cohen's *d* = −0.76; 95% CI: 24.79
to 2.13, $t_{16}$ = −2.520, *p* = 0.023), F4 (M: 31.14 ± 11.07%, NM: 40.04 ± 14.08%; Cohen's
*d* = −0.70; 95% CI: 16.88 to 0.91, $t_{16}$ = −2.364, *p* = 0.031), and F8 regions (M: 31.12 ± 9.60%,
NM: 43.14 ± 16.72%; Cohen's *d* = −0.88; 95% CI: 22.55 to 1.47, $t_{16}$ = −2.417, *p* = 0.028).

In addition, compared with no music intervention, listening to self-selected music
during the warm-up significantly downregulated θ/β in the F3 (M: 2.41 ± 2.65, NM:
5.04 ± 4.01; Cohen's *d* = −0.77; 95% CI: −4.37 to −0.86, $t_{16}$ = −3.167, *p* = 0.006) and O1
regions (M: 2.39 ± 1.77, NM: 11.63 ± 17.38; Cohen's *d* = −0.74; 95% CI: −17.97 to −0.48,
$t_{16}$ = −2.238, *p* = 0.040) and (θ+α)/(α+β) in the F3 region (M: 1.27 ± 0.41, NM: 2.09 ± 0.99;
Cohen's *d* = −1.07; 95% CI: −1.27 to −0.36, $t_{16}$ = −3.828, *p* = 0.001).

**Table 2.** Descriptions of EEG signals in music intervention and no music intervention trails.

| Brain Regions | δ (%) | | θ (%) | | α (%) | | β (%) | | θ/β | | (θ+α)/(α+β) | |
|---|---|---|---|---|---|---|---|---|---|---|---|---|
| | M | NM | M | NM | M | NM | M | NM | M | NM | M | NM |
| FP1 | 40.89 ± 16.05 | 44.70 ± 18.69 | 21.17 ± 5.48 | 20.14 ± 3.64 | 22.75 ± 10.17 | 22.04 ± 15.02 | 15.17 ± 8.73 | 13.13 ± 10.25 | 1.96 ± 1.25 | 3.42 ± 4.54 | 1.24 ± 0.28 | 1.65 ± 1.30 |
| F3 | 35.77 ± 13.97 ** | 51.27 ± 17.84 | 23.25 ± 6.15 | 24.14 ± 7.45 | 24.55 ± 12.33 * | 16.41 ± 11.10 | 16.40 ± 11.81 ** | 8.17 ± 5.89 | 2.41 ± 2.65 ** | 5.04 ± 4.01 | 1.27 ± 0.41 ** | 2.09 ± 0.99 |
| C3 | 34.02 ± 11.68 | 41.53 ± 14.68 | 25.04 ± 5.74 | 25.41 ± 6.32 | 28.72 ± 12.44 * | 23.74 ± 14.77 | 12.21 ± 6.68 | 9.31 ± 5.27 | 2.79 ± 1.90 | 3.56 ± 1.91 | 1.42 ± 0.38 | 1.61 ± 0.42 |
| P3 | 30.51 ± 10.22 * | 42.34 ± 19.78 | 25.44 ± 8.08 | 26.48 ± 10.77 | 31.74 ± 14.40 ** | 19.49 ± 12.92 | 12.30 ± 5.95 | 11.67 ± 15.16 | 2.51 ± 1.30 | 5.81 ± 9.16 | 1.44 ± 0.58 | 1.95 ± 1.52 |
| O1 | 36.31 ± 18.35 * | 49.77 ± 16.80 | 21.50 ± 6.93 | 25.59 ± 14.72 | 30.95 ± 17.29 ** | 18.35 ± 14.64 | 11.25 ± 4.65 ** | 6.29 ± 4.90 | 2.39 ± 1.77 * | 11.63 ± 17.38 | 1.37 ± 0.34 | 3.29 ± 3.85 |
| F7 | 32.56 ± 13.99 | 39.73 ± 16.13 | 25.90 ± 7.51 | 25.03 ± 8.63 | 25.15 ± 9.92 | 22.30 ± 13.38 | 16.38 ± 13.04 | 12.95 ± 10.46 | 2.75 ± 2.79 | 3.87 ± 5.02 | 1.37 ± 0.53 | 1.63 ± 0.94 |
| T3 | 33.55 ± 15.25 | 43.56 ± 15.34 | 24.28 ± 11.30 | 26.72 ± 11.29 | 26.85 ± 12.93 ** | 17.84 ± 9.26 | 15.30 ± 10.01 | 11.87 ± 8.94 | 3.28 ± 4.41 | 5.14 ± 6.45 | 1.55 ± 1.09 | 1.90 ± 1.24 |
| T5 | 27.50 ± 13.55 | 37.01 ± 17.89 | 18.24 ± 5.17 | 19.86 ± 3.85 | 35.94 ± 14.69 | 30.64 ± 15.43 | 18.28 ± 8.35 * | 12.48 ± 7.49 | 1.49 ± 1.55 | 3.01 ± 3.54 | 1.08 ± 0.35 | 1.48 ± 1.04 |
| FP2 | 38.24 ± 13.03 | 45.82 ± 15.82 | 22.58 ± 7.36 | 20.75 ± 5.10 | 28.48 ± 13.36 * | 21.26 ± 11.76 | 10.70 ± 5.60 | 12.16 ± 5.82 | 2.76 ± 1.92 | 2.71 ± 3.54 | 1.44 ± 0.46 | 1.60 ± 1.20 |
| F4 | 31.14 ± 11.07 * | 40.04 ± 14.08 | 24.80 ± 7.29 | 24.53 ± 5.64 | 32.40 ± 12.24 * | 24.78 ± 13.55 | 11.66 ± 6.49 | 10.65 ± 4.87 | 2.81 ± 1.89 | 2.78 ± 1.36 | 1.43 ± 0.48 | 1.51 ± 0.36 |
| C4 | 34.00 ± 12.54 | 38.37 ± 12.10 | 25.80 ± 7.13 | 25.80 ± 5.33 | 31.11 ± 13.54 | 24.68 ± 13.75 | 9.088 ± 4.85 | 11.13 ± 7.65 | 3.70 ± 2.21 | 3.30 ± 2.26 | 1.57 ± 0.48 | 1.55 ± 0.45 |
| P4 | 29.51 ± 13.29 | 35.31 ± 11.43 | 26.80 ± 12.00 | 25.82 ± 7.50 | 32.95 ± 13.64 | 27.52 ± 13.56 | 10.68 ± 7.04 | 11.32 ± 5.77 | 4.34 ± 6.46 | 2.91 ± 1.96 | 1.74 ± 1.49 | 1.49 ± 0.42 |
| O2 | 46.47 ± 12.29 | 47.69 ± 16.66 | 24.06 ± 7.82 | 24.06 ± 7.82 | 16.35 ± 4.92 | 15.18 ± 9.15 | 14.05 ± 5.20 | 13.06 ± 10.78 | 1.87 ± 0.78 | 2.89 ± 2.30 | 1.36 ± 0.32 | 1.66 ± 0.78 |
| F8 | 31.12 ± 9.60 * | 43.14 ± 16.72 | 25.90 ± 7.00 | 22.56 ± 5.22 | 29.94 ± 12.27 | 23.06 ± 13.20 | 13.01 ± 5.64 | 11.23 ± 5.18 | 2.54 ± 1.66 | 3.12 ± 4.13 | 1.40 ± 0.47 | 1.55 ± 0.75 |
| T4 | 29.94 ± 13.83 | 34.82 ± 12.00 | 23.10 ± 6.27 | 24.61 ± 9.67 | 32.70 ± 13.68 | 27.09 ± 10.14 | 14.25 ± 8.46 | 13.42 ± 6.78 | 2.53 ± 2.29 | 3.16 ± 5.27 | 1.32 ± 0.47 | 1.44 ± 0.77 |
| T6 | 28.27 ± 16.06 | 34.37 ± 16.26 | 19.60 ± 5.40 | 19.24 ± 7.38 | 39.39 ± 16.70 | 34.32 ± 15.94 | 12.74 ± 5.46 | 12.08 ± 6.07 | 1.92 ± 1.24 | 2.22 ± 1.76 | 1.23 ± 0.30 | 1.25 ± 0.32 |

Note: M, music intervention; NM, no music intervention; Fp1, left frontal poel; Fp2, right frontal poel; F3, left frontal; F4, right frontal; C3, left central; C4, right central; P3, left parietal; P4, right parietal; O1, left occipital; O2, right occipital; F7, left anterior temporal; F8, right anterior temporal; T3, left mid-temporal; T4, right mid-temporal; T5, left posterior temporal; T6, right posterior temporal region. * $p < 0.05$, ** $p < 0.01$.

## 4. Discussion

The aim of this study was to explore the effects of listening to self-selected music during a warm-up session on brain wave synchronization/desynchronization and Wingate test performance in healthy young men. The results showed that listening to self-selected music during a warm-up significantly increased peak power and mean power in the Wingate test. In addition, listening to self-selected music during a warm-up also upregulated the $\alpha$ energy percentage in the F3, C3, P3, O1, T3, F4, and FP2 regions, the $\beta$ energy percentage in the F3, O1, and T5 regions, and downregulated the $\delta$ energy percentage in the F3, P3, O1, F4, and F8 regions, $\theta/\beta$ in the F3 region and O1 regions, and $(\theta+\alpha)/(\alpha+\beta)$ in the F3 region compared to no music intervention.

### 4.1. Effects of Self-Selected Music on Anaerobic Performance

The results of this study showed that listening to self-selected music during a warm-up session improved anaerobic performance, mainly in terms of peak power and mean power of the Wingate test, which was consistent with the results of previous studies. For example, Eliadim et al. [8] showed that the peak power in the Wingate test was significantly higher in the music intervention group compared to the no music intervention group. In addition, Stork et al. [33] reported that both peak power and mean power were positively influenced by music. Furthermore, Isik et al. [34] found that compared with conditions without music intervention, motivational music significantly improved performance on the Wingate test, including maximum anaerobic power, maximum anaerobic capacity, relative anaerobic power, and relative anaerobic capacity. Moreover, Smirmaul et al. [7] showed that pre-task music had an ergogenic effect on shorter and predominantly anaerobic tasks such as Wingate test, grip strength, and short-duration sports.

The brain plays an important role in the neurophysiological mechanisms that mediate music interventions. Previous studies have confirmed that the response to music interventions is associated with the temporal, frontal, and insular cortex, which is expressed through changes in brain waves [35,36]. Changes in EEG signals (such as $\alpha$, $\beta$, $\theta$, and $\delta$) and derived indicators (such as $\theta/\beta$ and $(\theta+\alpha)/(\alpha+\beta)$) are associated with arousal levels, cortical excitability, and central fatigue. Proper activation reflects the arousal state of the brain, while over-activation reflects the over-excited state of the brain [37,38]. We next explored the possible mechanisms by which listening to self-selected music during a warm-up improved anaerobic performance by conducting an EEG after the music intervention. Our results showed that listening to self-selected music during a warm-up significantly upregulated the $\alpha$ energy percentage in the F3, C3, P3, O1, T3, F4, and FP2 regions and the $\beta$ energy percentage in the F3, O1, and T5 regions, indicating that listening to self-selected music during a warm-up increased the arousal level and brain excitability. The increase in $\alpha$ waves indicates that the firing rate of the central nervous system tends to be coordinated and synchronized, which reflects a stable trend of EEG changes [38]. In addition, $\beta$ waves are related to the degree of activation of nerve cells in the cerebral cortex and are the main form of expression of electrical activity when the cerebral cortex is in a state of tension and excitement, especially active thinking, focused, and high alert [39]. The functional organization of the brain dynamically regulates the connections between different regions through rapid neuronal activity changes over short periods of time in response to a variety of tasks [40]. A previous study has shown that listening to positive, emotionally arousing music enhanced interhemispheric coherence and activated frontal lobe messaging [41]. This activation is thought to reflect working memory for emotional and environmental perception, which is associated with the medial frontal lobe [42]. Moreover, the frontal lobes, especially the prefrontal cortex, are associated with executive functions and advanced cognitive processes [43]. Furthermore, behavioral types are correlated with rapid brain oscillations in different brain regions. For instance, emotion is associated with $\alpha$ oscillations in the frontal lobe, whereas visualspatial attention is associated with $\alpha$ oscillations in the parietal lobe. Focused internal attention is associated with $\alpha$ oscillations in the right temporal lobe, and object processing is associated with $\beta$ oscillations in the temporal and parietal

lobes [44]. In addition, our results showed that listening to self-selected music during a warm-up session significantly downregulated the δ energy percentage in the F3, P3, O1, F4, and F8 regions, but had no significant effect on the θ energy percentage. When the body is in a deep sleep or relaxed state, δ and θ waves are the main active brain waves. However, when the body is in an excited state, the degree of cortical inhibition is diminished and the power spectrum energy of δ and θ waves will be significantly decreased [45].

Based on these results, we found that the effects of the musical intervention on the cerebral cortex were mainly concentrated in the left hemisphere (e.g., left frontal, left central, left parietal, left occipital, etc.), which may be related to the fact that the subjects were right-handed. Our results were in agreement with the study of Altenmüller et al. [46], showing that for right-handed subjects, brain activation patterns during listening to music showed a highly significant lateralization effect; positive emotional attributions were accompanied by an increase in left temporal activation. Taken together, we may conclude that listening to self-selected music during a warm-up can significantly enhance the excitability of the cerebral cortex, especially the left hemisphere, which may account for the increase in peak power and mean power in the Wingate test.

### 4.2. Effects of Self-Selected Music on Fatigue

In this study, listening to self-selected music during a warm-up had no significant effect on fatigue, HR, and RPE after the Wingate test. When exercise leads to a deepening of central fatigue, the inhibitory progresses in the cerebral cortex are more dominant and the excitatory progresses are at a disadvantage [45,47]. The $θ/β$ and $(θ+α)/(α+β)$ levels are commonly used to determine fatigue, and increases in $θ/β$ and $(θ+α)/(α+β)$ imply an inhibition of arousal levels and the production of central fatigue [30,31,48]. EEG data showed that listening to self-selected music during a warm-up significantly downregulated the $θ/β$ levels in the F3 and O1 regions, and $(θ+α)/(α+β)$ levels in the F3 region compared to no music intervention. However, since C3 and C4 are the most representative regions in the sensorimotor area of the cerebral cortex [31], reduction in $θ/β$ and $(θ+α)/(α+β)$ levels in the F3 and O1 regions did not result in a reduction in exercise-induced fatigue, indicating that listening to self-selected music during a warm-up did not reverse exercise-induced fatigue, which was consistent with the results of Jarraya et al. [49], showing that listening to music during a warm-up had a beneficial effect on short-term supramaximal performances, while FI, HR, and RPE during the Wingate test were not affected by music. In addition, Urakawa et al. [50] found no significant difference in HR between the music intervention and no music intervention, suggesting the limited effect of pre-exercise music intervention on HR.

The effect of music on fatigue may be related to the intensity of the exercise performed after the music intervention. Potteiger et al. [12] reported that during moderate-intensity exercise, self-selected music intervention resulted in a reduced peripheral, central, and overall RPE compared to no music intervention. However, when the intensity of exercise rises to the extreme load, the effect of music falls far short of counteracting the fatigue associated with high-intensity exercise, which may be caused by the combined effect of mechanical and chemical stimulation received by the receptors of muscles, heart, and lungs during exercise [51]. Furthermore, although FI is directly related to the power values obtained in the Wingate test, it is more dependent on the ratio of minimum power to peak power [10]. Therefore, the enhancing effect of music on anaerobic performance may be evident at the beginning of the exercise and diminish as exercise progresses [8,52], making the effect of pre-exercise music interventions limited in improving fatigue and RPE. To sum up, listening to self-selected music during a warm-up cannot simultaneously improve anaerobic performance and reduce exercise fatigue.

### 4.3. Limitations

This study has some limitations that should be considered. First, only men were recruited, and the sample size of participants was small. In addition, only the acute effects

of self-selected music on cortical activation and anaerobic performance were examined. Therefore, future studies with larger sample sizes and implementation of longer term interventions are needed to examine and confirm the observations made in this study and to explore the long-term effects of self-selected music intervention. Second, we did not record endocrinological parameters, which could have contributed to elucidating our findings better. Third, the study relied on individuals' self-selected music during the warm-up. This introduced variability in music preferences, genres, and tempo, which may influence the psychological and physiological responses differently. Standardizing the music selection process or categorizing participants based on their musical preferences could provide more consistent and reliable results. Furthermore, Karageorghis et al. [53] showed that volume may affect the perception of music during exercise. The volume of music in this study was controlled by the participants themselves, rather than set to a fixed level to fit their habits of listening to music during exercise, which may have had some effect on the results. Finally, although subjects with more than one year of training experience were not intentionally selected during the recruitment process, the average training experience of subjects in this study was 3.65 years and the level of training experience may influence the results.

## 5. Conclusions

This study showed that listening to self-selected music during a warm-up session enhances excitability in the left hemisphere cerebral cortex, specifically the left frontal, left central, left parietal, and left occipital regions, by upregulating the $\alpha$ and $\beta$ energy percentages and downregulating the $\delta$ energy percentage, which may represent a potential mechanism by which listening to self-selected music during a warm-up improves anaerobic performance in healthy young men.

**Author Contributions:** Conceptualization, L.Y. and S.Z.; methodology, S.Z. and J.Y.; software, S.Z.; validation, J.Y., X.T. and L.D.; formal analysis, S.Z.; investigation, S.Z., J.Y., X.T., L.D., X.L., Y.L. and X.H.; resources, L.Y.; data curation, X.H.; writing—original draft preparation, S.Z. and J.Y.; writing—review and editing, L.Y.; visualization, X.H.; supervision, L.Y.; project administration, L.Y.; funding acquisition, L.Y. and J.Y. All authors have read and agreed to the published version of the manuscript.

**Funding:** This research was funded by the National Key R&D Program of China, grant number 2022YFC3600201, Chinese Universities Scientific Fund, grant number 2021QN001, 2022QN015, and Graduate Students' Innovative Scientific Research Program of Beijing Sport University, grant number 20221059.

**Institutional Review Board Statement:** The study was conducted in accordance with the Declaration of Helsinki, and approved by the Ethics Committee of Beijing Sport University (protocol code 2022115H, 2022/05/16).

**Informed Consent Statement:** Informed consent was obtained from all subjects involved in the study.

**Data Availability Statement:** The datasets generated during and/or analyzed during the present study are available from the corresponding author upon reasonable request.

**Conflicts of Interest:** The authors declare no conflict of interest.

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
