# Peer review of "Listening to Self-Selected Music during Warm-Up Improves Anaerobic Performance through Enhancement of the Excitability of the Cerebral Cortex"

_applsci, doi:10.3390/app13127180_

Round 1
Reviewer 1 Report
Dear authors,
Thank you for your work. I’ve some significant considerations for you, and I hope it helps to improve your paper.
Introduction
Line 57: Please, correct the double space there.
Lines 78: Before talking about music, it might be good to introduce the phrase ‘the alpha brain wave has more activation in intermittent noisy environments than in completely silent environments (Domingos et al., 2021)1’. After that, you go to the music part. I believe this is the best option.
Methods
From what I understand, participants could do 4 Wingate or 4 EEGs because they were randomly assigned to it, or they always did an exact number of Wingate and EEG? Please clarify.
Line 111: What does this ‘more than one year of training experience’ mean? That you had subjects with 5 or more years and others with only 1? And experience in what?
EEG recording: If subjects close their eyes, the alpha will increase, and it will be unfair to compare with other activities (Wingate in this case).
Also, I do not understand the point of having a measure for the resting state and another utterly different measure for Wingate without EEG. What do you want to show or prove? How can a resting state (EEG measure) help understand possible improvements in anaerobic performance?
Statistical Analysis: Using parametric statistics for brain measures doesn’t sound correct. Are you sure that normality was reached with the EEG parameters? Moreover, could you provide more explanation about all the deartifaction process. This is the most critical step because your results will be biased if you have clean and correct data.
Results
In this section, the subtitles seem to be the same. Correct it, please.
Discussion
The first paragraph of your discussion sounds like they only did Wingate, and they had an EEG cap while doing Wingate. Is that true? If so, I had understood everything wrong in your study, and thus the methods need to be more explicit. If not, you must rewrite this paragraph and separate the results obtained in the resting state and Wingate without the cap. Furthermore, I do not remember that the objective or hypothesis of your study was to up- and down-regulate the ‘energy percentage’ (which is inadequate to say so).
Only in lines 291-293, I did understand the meaning of doing EEG in a resting state…
There are many more limitations that you should definitely include.
Conclusions
With that sample size and without a power sample, you can infer with certainty that conclusion. You must be more cautious about the way you express your conclusion.
1Domingos C, da Silva Caldeira H, Miranda M, Melício F, Rosa AC, Pereira JG. The Influence of Noise in the Neurofeedback Training Sessions in Student Athletes. International Journal of Environmental Research and Public Health. 2021; 18(24):13223. https://doi.org/10.3390/ijerph182413223
Minor.
Author Response
Point 1: Line 57: Please, correct the double space there.
Response 1: thank you very much for this important comment.
We have corrected the double space. (Please see the revised manuscript at line 58 of page 2).
Point 2: Lines 78: Before talking about music, it might be good to introduce the phrase‘the alpha brain wave has more activation in intermittent noisy environments than in completely silent environments (Domingos et al., 2021)1’. After that, you go to the music part. I believe this is the best option.
Response 2: thank you very much for this important comment.
The content “A previous study showed that the α wave has more activation in intermittent noisy environments than in completely silent environments [24].” has been added. (Please see the revised manuscript at line 97-99 of page 3).
Point 3: Methods: From what I understand, participants could do 4 Wingate or 4 EEGs because they were randomly assigned to it, or they always did an exact number of Wingate and EEG? Please clarify.
Response 3: thank you very much for this important comment.
The procedures were: subjects were randomized to (1) EEG test after a 10-minute warm-up session without music intervention; (2) Wingate test after a 10-minute warm-up session without music intervention; (3) EEG test after a 10-minute warm-up session with music intervention; and (4) Wingate test after a 10-minute warm-up session with music intervention.
Point 4: Line 111: What does this‘more than one year of training experience’mean? That you had subjects with 5 or more years and others with only 1? And experience in what?
Response 4: thank you very much for this important comment.
The content “Seventeen healthy young men [age: 21.31 ± 1.01 years; height: 179.35 ± 4.36 cm; weight: 74.65 ± 7.05 kg; body mass index (BMI): 23.18 ± 1.63 kg/m2] with more than one year of sports training experience (3.65 ± 0.70 years) volunteered to participate in this study.” has been modified. (Please see the revised manuscript at line 133-136 of page 4).
Point 5: Statistical Analysis: Using parametric statistics for brain measures doesn’t sound correct. Are you sure that normality was reached with the EEG parameters? Moreover, could you provide more explanation about all the deartifaction process. This is the most critical step because your results will be biased if you have clean and correct data.
Response 5: thank you very much for this important comment.
Actually, we used non-parametric tests to compare the non-normal data.
The content “For non-normal data, non-parametric tests were used.” has been added. (Please see the revised manuscript at line 213-214 of page 6).
The content “All EEG signals were preprocessed in MATLAB to remove artifacts and noise. Ocular and cardiac artifacts were removed using the independent components analysis (ICA) from the EEGLAB toolbox . Subsequently, all signals were digitally filtered with a band pass Butterworth filter between 0.5 and 50 Hz.” has been added. (Please see the revised manuscript at line 200-204 of page 6).
Point 6: Results: In this section, the subtitles seem to be the same. Correct it, please.
Response 6: thank you very much for this important comment.
We have corrected the subtitles. (Please see the revised manuscript at line 234 and 245 of page 7-8).
Point 7: Discussion: The first paragraph of your discussion sounds like they only did Wingate, and they had an EEG cap while doing Wingate. Is that true? If so, I had understood everything wrong in your study, and thus the methods need to be more explicit. If not, you must rewrite this paragraph and separate the results obtained in the resting state and Wingate without the cap. Furthermore, I do not remember that the objective or hypothesis of your study was to up- and down-regulate the‘energy percentage’(which is inadequate to say so).
Response 7: thank you very much for this important comment.
The procedures were: subjects were randomized to (1) EEG test after a 10-minute warm-up session without music intervention; (2) Wingate test after a 10-minute warm-up session without music intervention; (3) EEG test after a 10-minute warm-up session with music intervention; and (4) Wingate test after a 10-minute warm-up session with music intervention.
The content “The results showed that listening to self-selected music during warm-up significantly increased peak power and mean power in the Wingate test. In addition, listening to self-selected music during warm-up also upregulated α energy percentage in the F3, C3, P3, O1, T3, F4, and FP2 regions, β energy percentage in the F3, O1, and T5 regions, and downregulated δ energy percentage in the F3, P3, O1, F4, and F8 regions, θ/β in the F3 region and O1 regions, and (θ+α)/(α+β) in the F3 region compared to no music intervention.” has been modified. (Please see the revised manuscript at line 285-291 of page 10).
Point 8: There are many more limitations that you should definitely include.
Response 8: thank you very much for this important comment.
The content “This study has some limitations that should be considered. First, only men were recruited, and the sample size of participants was small. In addition, only the acute effects of self-selected music on cortical activation and anaerobic performance were examined. Therefore, future studies with larger sample sizes and implementation of longer term of intervention are needed to examine and confirm the observations made in this study and to explore the long-term effects of self-selected music intervention. Second, we did not record endocrinological parameters, which could have contributed to elucidating our findings better. Third, the study relied on individuals' self-selected music during warm-up. This introduced variability in music preferences, genres, and tempo, which may influence the psychological and physiological responses differently. Standardizing the music selection process or categorizing participants based on their musical preferences could provide more consistent and reliable results. Furthermore, Karageorghis et al. [53] showed that volume may affect the perception of music during exercise. The volume of music in this study was controlled by the participants themselves, rather than set to a fixed level to fit their habits of listening to music during exercise, which may have had some effect on the results. Finally, although subjects with more than one year of training experience were not intentionally selected during the recruitment process, the average training experience of subjects in this study was 3.65 years, the level of training experience may influence the results.” has been modified. (Please see the revised manuscript at line 390-408 of page 12-13).
Point 9: Conclusions: With that sample size and without a power sample, you can infer with certainty that conclusion. You must be more cautious about the way you express your conclusion.
Response 9: Thank you very much for your valuable comments and suggestions on our manuscript.
The content “This study showed that listening to self-selected music during warm-up is of great promise to upregulate α energy percentage in the left frontal, left central, left parietal, left occipital, left mid-temporal, right frontal, and right frontal poel, β energy percentage in the left frontal, left occipital, and left posterior temporal, and downregulate δ energy percentage in the left frontal, left parietal, left occipital, right mid-temporal, and right anterior temporal, θ/β in the left frontal and left occipital, and (θ+α)/(α+β) in the left frontal. Furthermore, listening to self-selected music during warm-up enhances the excitability of the cortical cortex by upregulating α and β energy percentages and downregulating δ energy percentage may represent a potential mechanism by which listening to self-selected music during warm-up improves anaerobic performance in healthy young men.” has been modified. (Please see the revised manuscript at line 410-420 of page 13).

Reviewer 2 Report
THEORETICAL PART:
1) Imprecise and incorrect definition of EEG. It is not a synthesis of the electrical activity of a population of neurons.
2) EEG waves can be systematized in several ways. In muscle research, it would be useful to distinguish SMR waves, which the authors did not take into account. Why?
3) Too little literature and references to other studies in the introduction. Among other things, the authors should have addressed the topic of music therapy, which has been very extensively described in neuroimaging studies of the brain.
4) They did not explain why music intervention activates frontal, shadow and temporal areas. The authors referred to studies without a more thorough description. This is important because each has a different distribution of waves i.e. that in one Alpha wave stimulation will be desirable in another not. The authors made an oversimplification.
METHODOLOGY
1) It is not possible to study the effect of anything on EEG signals; rather, the purpose of the study should be to study the effect on brain wave synchronization/desynchronization
2) The sample selection did not describe whether epilepsy or brain injury was an exclusion criterion. The use of the word ''healthy'' is not precise here
3) It is not clear whether subjects with more than one year of training experience were intentionally selected? Could the length of experience affect the results?
4) Why weren't more in-depth analyses made of, for example, the fact that most of the significant differences were in the left hemisphere (were the participants right-handed, was interhemispheric bias checked?)
5) There is also a lack of conclusions related to the functional specificity of the regions that were statistically significantly stimulated (conclusions were presented in a very cursory manner).
Author Response
Point 1: Imprecise and incorrect definition of EEG. It is not a synthesis of the electrical activity of a population of neurons.
Response 1: thank you very much for this important comment.
The content “Electroencephalogram (EEG) records the electrical discharges of a group of neurons in the brain during synchronous activation through electrodes attached to the scalp [20,21]. EEG power reflects the number of neurons that electrical discharge synchronously [22].” has been modified. (Please see the revised manuscript at line 91-94 of page 3).
Point 2: EEG waves can be systematized in several ways. In muscle research, it would be useful to distinguish SMR waves, which the authors did not take into account. Why?
Response 2: thank you very much for this important comment.
Since SMR waves are mainly related to the idle state of the motor cortex, in this study, we consider that music intervention can also affect other areas of the cerebral cortex.
Point 3: Too little literature and references to other studies in the introduction. Among other things, the authors should have addressed the topic of music therapy, which has been very extensively described in neuroimaging studies of the brain.
Response 3: thank you very much for this important comment.
The content “In addition, music therapy has a wide range of applications in the field of nursing and rehabilitation, providing sufficient evidence for the impact of music intervention on brain neuroimaging. A previous study showed that listening to music had a positive effect on the recovery of language function after stroke, and the increased activation of the left frontal aslant tract mediated structural neuroplastic changes in the language network [17]. Brattico et al. [18] used functional magnetic resonance imaging (fMRI) to investigate brain responses to sad and positive music with or without lyrics. They found that sad music with lyrics recruited more brain activity in several regions, including the caudate nucleus, parahippocampal gyrus, inferior frontal gyrus, amygdala, precentral gyrus, putamen, insula, and auditory cortex. On the other hand, positive music with lyrics was activated in the limbic system and certain regions of the right inferior frontal gyrus. Whitehead et al. [19] found differences in the response to sound and music stimuli in different regions of the temporal lobe of the brain, specifically, the superior temporal sulcus and gyrus were more active in response to sound stimuli, whereas planum polare and temporal were more active in response to music stimuli.” has been modified. (Please see the revised manuscript at line 75-90 of page 2-3).
Point 4: They did not explain why music intervention activates frontal, shadow and temporal areas. The authors referred to studies without a more thorough description. This is important because each has a different distribution of waves i.e. that in one Alpha wave stimulation will be desirable in another not. The authors made an oversimplification.
Response 4: thank you very much for this important comment.
The content “The functional organization of the brain dynamically regulates the connections between different regions through rapid neuronal activity changes over short periods of time in response to variety of tasks [40]. A previous study has shown that listening to positive, emotionally arousing music enhanced interhemispheric coherence and activated frontal lobe messaging [41]. This activation is thought to reflect working memory for emotional and environmental perception, which is associated with the medial frontal lobe [42]. Moreover, the frontal lobes, especially the prefrontal cortex, are associated with executive functions and advanced cognitive processes [43]. Furthermore, behavioral types are correlated with rapid brain oscillations in different brain regions. For instance, emotion is associated with α oscillations in the frontal lobe, whereas visualspatial attention is associated with α oscillations in the parietal lobe. Focused internal attention is associated with α oscillations in the right temporal lobe, and object processing is associated with β oscillations in the temporal and parietal lobes [44].” has been modified. (Please see the revised manuscript at line 324-337 of page 11).
Point 5: It is not possible to study the effect of anything on EEG signals; rather, the purpose of the study should be to study the effect on brain wave synchronization/desynchronization.
Response 5: thank you very much for this important comment.
The content “The study investigated the effects of listening to self-selected music during warm-up on brain wave synchronization/desynchronization and Wingate test performance.” has been modified. (Please see the revised manuscript at line 17-18 of page 1).
The content “In the present study, we aimed to investigate the effects of listening to self-selected music during warm-up on brain wave synchronization/desynchronization and Wingate test performance in healthy young men in order to explore possible mechanisms by which music intervention improves anaerobic performance.” has been modified. (Please see the revised manuscript at line 116-120 of page 4).
The content “The aim of this study was to explore the effects of listening to self-selected music during warm-up on brain wave synchronization/desynchronization and Wingate test performance in healthy young men. ” has been modified. (Please see the revised manuscript at line 283-285 of page 10).
Point 6: The sample selection did not describe whether epilepsy or brain injury was an exclusion criterion. The use of the word ''healthy'' is not precise here.
Response 6: thank you very much for this important comment.
Epilepsy and brain injury can have significant effects on brain function and can manifest in altered EEG activity. In addition, there are various other factors that researchers might consider when defining the term "healthy" in the context of sample selection for an EEG study. Therefore, we provided additional information about the sample selection criteria. The content “None were taking drugs or had antecedents of neurological or psychopathological conditions.” has been added. (Please see the revised manuscript at line 138-139 of page 4).
Point 7: It is not clear whether subjects with more than one year of training experience were intentionally selected? Could the length of experience affect the results?
Response 7: thank you very much for this important comment.
Subjects with more than one year of training experience were not intentionally selected during the recruitment process, we just described the physical characteristics of subjects. The length of experience may affect the results, so we acknowledged this in the limitations. (Please see the revised manuscript at line 405-408 of page 13).
Point 8: Why weren't more in-depth analyses made of, for example, the fact that most of the significant differences were in the left hemisphere (were the participants right-handed, was interhemispheric bias checked?)
Response 8: thank you very much for this important comment.
The content “Based on these results, we found that the effects of the musical intervention on the cerebral cortex were mainly concentrated in the left hemisphere (e.g. left frontal, left central, left parietal, left occipital, etc.), which may be related to the fact that the subjects were right-handed. Our results were in agreement with the study of Altenmüller et al. [46], showing that for right-handed subjects, brain activation patterns during listening to music showed a highly significant lateralization effect: positive emotional attributions were accompanied by an increase in left temporal activation. Taken together, we may conclude that listening to self-selected music during warm-up can significantly enhance the excitability of the cerebral cortex, especially the left hemisphere, which may account for the increase in peak power and mean power in the Wingate test.” has been added. (Please see the revised manuscript at line 344-354 of page 11).
Point 9: There is also a lack of conclusions related to the functional specificity of the regions that were statistically significantly stimulated (conclusions were presented in a very cursory manner).
Response 9: thank you very much for this important comment.
The content “This study showed that listening to self-selected music during warm-up is of great promise to upregulate α energy percentage in the left frontal, left central, left parietal, left occipital, left mid-temporal, right frontal, and right frontal poel, β energy percentage in the left frontal, left occipital, and left posterior temporal, and downregulate δ energy percentage in the left frontal, left parietal, left occipital, right mid-temporal, and right anterior temporal, θ/β in the left frontal and left occipital, and (θ+α)/(α+β) in the left frontal. Furthermore, listening to self-selected music during warm-up enhances the excitability of the cortical cortex by upregulating α and β energy percentages and downregulating δ energy percentage may represent a potential mechanism by which listening to self-selected music during warm-up improves anaerobic performance in healthy young men.” has been modified. (Please see the revised manuscript at line 410-420 of page 13).

Round 2
Reviewer 1 Report
Dear authors,
Before starting to answer your 'modifications', I'd like to say that, personally, using always the same comment: "Thank you very much for this important comment.". Means nothing to me. It feels like you were forced to write this so the reviewers would be pleased. We are human, too. If it is not sincere, avoid it.
Back to the main point. I'm delighted that you mentioned the lines of your modifications. It helps a lot!!
In point 3, that's not the answer I was looking for, but that's not your fault. After re-checking the methods, I just realized that I missed that it was a randomized CROSSOVER design! Now I understand the design of your study, despite the small sample size.
Point 4 - We still don't understand whether the participants are professional athletes. Doing sports can mean a lot of things. Federated or not federated, for example? Because if they're not federated, for me, they're not doing sport, but practicing physical exercise (such as gym, for example).
Point 5 - "Statistical Analysis: Using parametric statistics for brain measures doesn't sound correct. Are you sure that normality was reached with the EEG parameters? Moreover, could you provide more explanation about all the deartifaction process. This is the most critical step because your results will be biased if you have clean and correct data."
The answer to his suggestion proves nothing. Because you clearly state, "The significance of differences in RPE, HR, EEG signals and 200 Wingate test was assessed by paired sample t-test". Besides, adding that you used non-parametric statistics for non-normal data means nothing again. Which non parametric for which variables?
It is imperative to show proper analysis. Convince me that you did it correctly this time.
Minor.
Author Response
Point 4 - We still don't understand whether the participants are professional athletes. Doing sports can mean a lot of things. Federated or not federated, for example? Because if they're not federated, for me, they're not doing sport, but practicing physical exercise (such as gym, for example).
Response 4: Our participants are not professional athletes and they are not federated. We agree with your opinion that they are practicing physical exercise. Therefore, we removed the description of “training experience” in order not to cause confusion. (Please see the revised manuscript at line 133-135 of page 4).
Point 5 - "Statistical Analysis: Using parametric statistics for brain measures doesn't sound correct. Are you sure that normality was reached with the EEG parameters? Moreover, could you provide more explanation about all the deartifaction process. This is the most critical step because your results will be biased if you have clean and correct data."
The answer to his suggestion proves nothing. Because you clearly state, "The significance of differences in RPE, HR, EEG signals and 200 Wingate test was assessed by paired sample t-test". Besides, adding that you used non-parametric statistics for non-normal data means nothing again. Which non parametric for which variables?
Response 5: In this study, we used paired samples t-test to compare the normal data and non-parametric test to compare non-normal data. The content “Normality was checked using the Shapiro-Wilk test. After confirming the normality of the data, paired samples t-test was used. For non-normal data, non-parametric tests were used.” has been modified. (Please see the revised manuscript at line 210-212 of page 6).

Reviewer 2 Report
Thank you for completing the manuscript. Before publication, please improve the last suggestion, regarding conclusions.
Point 9: There is also a lack of conclusions related to the functional specificity of the regions that were statistically significantly stimulated (conclusions were presented in a very cursory manner).
It is not necessary to describe here where significant results were obtained, but what these areas are responsible for (functionally) and what this implies for the final sugetions/indications. It is important for another researchers.
Author Response
Point 9: There is also a lack of conclusions related to the functional specificity of the regions that were statistically significantly stimulated (conclusions were presented in a very cursory manner).
It is not necessary to describe here where significant results were obtained, but what these areas are responsible for (functionally) and what this implies for the final sugetions/indications. It is important for another researchers.
Response 9: thank you very much for this important comment.
The content “This study showed that listening to self-selected music during warm-up enhances excitability in the left hemisphere cerebral cortex, specifically the left frontal, left central, left parietal, and left occipital, by upregulating α and β energy percentages and downregulating δ energy percentage, which may represent a potential mechanism by which listening to self-selected music during warm-up improves anaerobic performance in healthy young men.” has been modified. (Please see the revised manuscript at line 409-414 of page 13).
